# Nanoparticles and Airway Epithelial Cells: Exploring the Impacts and Methodologies in Toxicity Assessment

**DOI:** 10.3390/ijms25147885

**Published:** 2024-07-18

**Authors:** Claire E. Lee, Fariba Rezaee

**Affiliations:** 1Department of Inflammation and Immunity, Lerner Research Institute, Cleveland Clinic Foundation, Cleveland, OH 44195, USA; leec23@ccf.org; 2Department of Cognitive Science, College of Arts and Sciences, Case Western Reserve University, Cleveland, OH 44106, USA; 3Center for Pediatric Pulmonary Medicine, Cleveland Clinic Children’s, Cleveland, OH 44195, USA

**Keywords:** airway epithelial cells, apical junctional complex, tight junction, adherens junction, nanoparticles, epithelial barrier dysfunction, permeability, inflammation, oxidative stress

## Abstract

The production of nanoparticles has recently surged due to their varied applications in the biomedical, pharmaceutical, textile, and electronic sectors. However, this rapid increase in nanoparticle manufacturing has raised concerns about environmental pollution, particularly its potential adverse effects on human health. Among the various concerns, inhalation exposure to nanoparticles poses significant risks, especially affecting the respiratory system. Airway epithelial cells play a crucial role as the primary defense against inhaled particulate matter and pathogens. Studies have shown that nanoparticles can disrupt the airway epithelial barrier, triggering inflammatory responses, generating reactive oxygen species, and compromising cell viability. However, our understanding of how different types of nanoparticles specifically impact the airway epithelial barrier remains limited. Both in vitro cell culture and in vivo murine models are commonly utilized to investigate nanoparticle-induced cellular responses and barrier dysfunction. This review discusses the methodologies frequently employed to assess nanoparticle toxicity and barrier disruption. Furthermore, we analyze and compare the distinct effects of various nanoparticle types on the airway epithelial barrier. By elucidating the diverse responses elicited by different nanoparticles, we aim to provide insights that can guide future research endeavors in assessing and mitigating the potential risks associated with nanoparticle exposure.

## 1. Introduction

Nanoparticle manufacturing has applications in numerous fields, such as agriculture, medicine, cosmetics, environmental monitoring, and drug development [1,2]. Notably, nanoparticles like titanium dioxide (TiO_2_) are used in food additives and coatings to preserve the shelf life of fruits [3,4,5]. Nanoparticles (NPs) typically range from 1 to 100 nm in diameter and exhibit varied physicochemical properties depending on their size, elemental composition, and specific crystal structure [6,7]. They can block UV radiation by absorbing, scattering, and reflecting both short and long wavelengths of ultraviolet light [8]. Nanoparticles have high surface-to-volume ratios, which enhances their thermal conductivity and catalytic activity [9]. Metal nanoparticles have become a popular material used for wound healing, tumor targeting, and other biomedical approaches [10]. Metal nanoparticles predominantly comprise pure metal and metal oxide nanoparticles. Pure metal nanoparticles include silver nanoparticles (AgNPs), gold nanoparticles (AuNPs), and cobalt nanoparticles (CoNPs). These compounds have enhanced thermodynamic stability and antimicrobial properties [11,12,13]. Similarly, metal oxide nanoparticles like TiO_2_-NPs, zinc oxide nanoparticles (ZnO-NPs), aluminum oxide nanoparticles (Al_2_O_3_-NPs), and copper oxide nanoparticles (CuO-NPs) are utilized to inhibit the transmission of pathogens [14]. Metal oxides also have optical properties such as UV absorption and photoluminescence [15]. However, inhalation of metal nanoparticles has been reported to cause detrimental effects such as lung inflammation and carcinogenesis [16]. The mechanisms causing these adverse responses involve the release of radical oxygen species and penetration of the airway barrier by metal ions. Additionally, nonmetal nanoparticles like silicon dioxide nanoparticles (SiO_2_-NPs) and graphene oxide nanoparticles (GONPs) contribute to airway hypersensitivity, fibrogenic responses, and oxidative stress [17].

Airway epithelial cells (AECs) play a crucial role in understanding the interactions between metal or nonmetal nanoparticles and the respiratory tract. These cells are the primary targets for pathogens and harmful particles via inhalation [18,19,20]. Alongside AECs, intercellular apical junctional complexes (AJCs) comprising tight junctions (TJs) and adherens junctions (AJs) are responsible for junction assembly/disassembly and interactions with nearby cells [21,22]. Together, the barrier function and mucociliary clearance presented by AECs, as well as the maintenance of barrier integrity by AJCs, serve as a dynamic defense structure [23]. Nanoparticles cause dysfunction and have a detrimental impact on the AJCs by increasing the permeability of the epithelial barrier [24,25] (Figure 1). Due to their small size, nanoparticles deposit deep in the lung, with increasing deposition efficiencies as their diameter decreases [26]. Common manifestations of nanoparticle exposure include fibrosis and chronic inflammation in addition to epithelial injury [27].

Despite the growing industry of nanomaterials, limited knowledge exists regarding nanoparticle toxicity and the associated risks to the respiratory tract. Specifically, there is a lack of comparative studies on different types of nanoparticles and their impact on the airway epithelial barrier. Past studies have used cell culture and murine models as in vitro and in vivo methods, respectively, to examine nanoparticle-induced inflammatory and cellular responses. Using primary or immortalized cell lines can provide insights into nanoparticle uptake and the functional activity of AECs [28,29]. Similarly, experiments on mice and rats can offer a more representative model for assessing structural and functional changes in the lung [30]. Both model types encounter challenges such as inconsistent cell morphology and differences in lung anatomy between mice/rats and humans [31,32]. However, they provide a valuable understanding of the effects of nanoparticles on the airway epithelial barrier at a cellular level. In this review, we discuss various cell culture and murine models that investigate the impact of common metal and nonmetal nanoparticles on AECs. We focus on the cellular responses of primary and immortalized cell lines, as well as mice and rat models. We reveal the specific mechanisms by which nanoparticles disrupt the airway epithelial barrier and compare the effects of various nanoparticle types. This insight will benefit future studies in selecting the appropriate experimental model based on the nanoparticle of interest.

## 2. Cell Culture Studies on Airway Cells

Cultured airway epithelial cells retain certain cell characteristics, which are valuable for studying structural and functional changes in the airway barrier induced by the inhalation of harmful particulate matter and inflammatory responses [33]. Cell culture models provide a simplified, controlled environment that allows different cell lines to be studied for nanoparticle toxicity in the epithelial airway (Table 1). Here, we focus on the well-studied 16HBE14o-, A549, Calu-3, and NHBE cell lines.

### 2.1. 16HBE14o- (16HBE) Human Bronchial Epithelial Cells

The 16HBE14o- (16HBE) cell line is a commonly used in vitro model for airway barrier properties and was derived from human bronchial epithelium immortalized with the SV40 plasmid [34,35]. The 16HBE cells have stable TJ morphology and have been shown to be suitable for assessing the airway epithelium’s permeability function [36,37]. TJs are located in the apical region of the AJC and create a “fence” between adjacent cells [38]. This “fence” maintains selective permeability for the proper exchange and paracellular transport of ions. When barrier disruption occurs, loss of permeability in the epithelial membrane causes uncontrolled, leaky airways [39]. In regard to barrier function, the exposure of 16HBE cells to TiO_2_-NPs has been shown to increase epithelial membrane permeability by decreasing transepithelial electrical resistance (TEER) [40]. For example, Lee et al. exposed 16HBE cells to 100 µg/mL of TiO_2_-NPs for 48 h [41]. They observed a significant decrease in TEER values, which shows barrier dysfunction caused by increased permeability of the epithelial cell monolayers. Immunofluorescent labeling also exhibited the disruption of TJ and AJ structures and reduced fluorescence intensity for cells exposed to TiO_2_-NPs. Conversely, the authors found no changes in TJ and AJ protein levels. This indicates that TiO_2_-NPs induced airway barrier disruption by facilitating the disassembly of TJ and AJ proteins rather than altering protein expression. In addition, due to the high reproducibility of 16HBE cells, this cell line has been widely used to examine the effect of nanoparticles on airway inflammatory responses and reactive oxygen species (ROS) production. For instance, it has been demonstrated that exposure of 16HBE cells to TiO_2_-NPs increased intracellular ROS levels at 1–100 μg/mL. However, the cell viability mainly decreased at concentrations above 10 μg/mL [42]. Likewise, another study reported that TiO_2_-NPs caused an increase in intracellular ROS production at concentrations of 15 and 37.5 μg/cm^2^ [43]. Bao et al. found that SiO_2_-NPs caused oxidative stress and promoted apoptosis in 16HBE cells [44]. The authors observed an increase in expression of Bax, a pro-apoptosis protein, and a decrease in Bcl-2 expression, an anti-apoptosis protein. Another study examined the impact of ZnO-NPs on the generation of ROS by utilizing 16HBE cells [45]. They showed that ZnO-NPs induced ROS generation and increased cell apoptosis by down-regulating the mRNA and protein expression of the anti-apoptosis protein Bcl-2. Mitochondrial membrane potential also decreased following exposure to ZnO-NPs. In general, ZnO-NPs have been shown to have the most cytotoxic impact on 16HBE cells compared to TiO_2_-NPs and SiO_2_-NPs [46,47].

### 2.2. A549 Adenocarcinoma (A549) Cells

The A549 adenocarcinoma cell line (A549) was first obtained from a type II pneumocyte lung tumor and is used to represent alveolar type II cells in the cell culture models [48]. As such, A549 cells are an appropriate model for studying the impact of nanoparticles on the peripheral lung region [49]. Like 16HBE cells, the A549 cell line has shown increased intracellular ROS levels and apoptosis caused by TiO_2_-NPs. Srivastava et al. exposed A549 cells to 5–50 μg/mL of TiO_2_-NPs for 6–24 h [50]. They found that 10–50 μg/mL of TiO_2_-NPs significantly increased ROS production at all time points. Additionally, they saw an increase in the expression of apoptotic markers, P^53^, P^21^, and caspase-3, which confirms the upregulation of apoptosis. Furthermore, Wang et al. used higher TiO_2_-NPs concentrations of 100–200 μg/mL on A549 cells [51]. They used quantitative real-time PCR (qRT-PCR) to measure the mRNA expression of apoptosis markers, caspase-3, and caspase-9. This study found a significant increase in the expression of both markers at all TiO_2_-NP concentrations. Previous studies have shown that other types of nanoparticles, such as AgNPs, ZnO-NPs, and SiO_2_-NPs, induce cytotoxicity in A549 cells as well. Chairuangkitti et al. assessed the adverse impact of AgNPs on A549 cells [52]. The authors found that A549 exposure to 100–200 μg/mL of AgNPs for 48 h led to an increase in intracellular ROS levels. Also, they observed a 50% reduction in cell viability after a 48 h exposure to 200 μg/mL of AgNPs. A study by Zhuo et al. examined how ZnO-NPs caused cellular damage to A549 cells [53]. They showed a significant decrease in cell viability at 15–40 μg/mL of ZnO-NPs for 24 h. Intracellular ROS production was seen to increase at 5–20 μg/mL concentrations, and this was thought to induce mitochondrial damage in A549 cells. Rafieepour et al. examined the effect of SiO_2_-NPs on A549 cells [54]. This study exposed A549 cells to 10–250 μg/mL of SiO_2_-NPs for 24 and 72 h. The investigators showed an increase in intracellular ROS generation for all SiO_2_-NP concentrations and a decrease in cell survival. Mitochondrial membrane potential was also shown to decrease, which is in agreement with the findings by Zhuo et al. [53]. Collectively, these studies show that the exposure of A549 cells to TiO_2_-NPs, AgNPs, ZnO-NPs, and SiO_2_-NPs triggers ROS production and apoptosis.

Interestingly, a study by Ma et al. revealed that A549 cells were more sensitive to 25 nm-sized TiO_2_-NPs than 16HBE cells [42]. The authors found that exposure to 1–100 μg/mL of TiO_2_-NPs caused significant cytotoxicity in A549 cells after 24 h. However, it took 48 h for the same cytotoxic increase to occur in the 16HBE. In a similar study, Guadagnini et al. exposed A549 and 16HBE cells to 3–75 µg/cm^2^ of TiO_2_-NPs and SiO_2_-NPs [43]. They showed that TiO_2_-NPs and SiO_2_-NPs generated cytotoxicity in A549 cells at lower doses and shorter exposure times than 16HBE cells.

A limitation of A549 cells is their lack of transepithelial resistance due to the absence of an intact TJ structure [55]. The TEER for A549 cells is generally less than 100 Ω×cm^2^ [56,57], making it difficult to study barrier function. However, A549 cells are regularly used as a single monolayer or co-culture with other epithelial cell lines to measure oxidative stress and cell viability, among other cellular responses. For instance, Braun et al. co-cultured A549 cells with the U937, a human monocytic cell line, to examine the effects of AgNPs [58]. The cultures were exposed to 5 µg/mL of AgNPs for 24 h. Compared to the A549 monoculture, the co-culture had no change in cell viability, implying that the presence of U937 attenuated cytotoxicity caused by AgNPs. Also, ROS production decreased in the co-culture group compared to just A549 exposure to AgNPs. Furthermore, Könczöl et al. evaluated the impact of metal-sulfate nanoparticles on A549 cells [59]. In this study, A549 cells were exposed to 50 and 100 μg/cm^2^ of calcium sulfate NPs (CaSO_4_-NPs), 1 and 10 μg/cm^2^ of zinc sulfate NPs (ZnSO_4_-NPs), or 100 μg/cm^2^ of lead sulfate NPs (PbSO_4_-NPs) for 24 h. They found that only ZnSO_4_-NPs induced concentration-dependent cytotoxicity. All the NPs were observed to cause oxidative stress, with ZnSO_4_-NPs showing the most intracellular ROS production. Genotoxicity assays were used to demonstrate DNA damage caused by ZnSO_4_-NPs, while PbSO_4_-NPs and CaSO_4_-NPs led to less DNA damage. The authors found that ZnSO_4_-NPs activated c-Jun N-terminal kinase (JNK), a regulator of apoptosis that responds to extracellular and intracellular stress. In another study, Stearns et al. exposed A549 to 40 μg/mL of TiO_2_-NPs for 3, 6, and 24 h [60]. At every time point, they observed membrane-bound vacuoles that contained aggregates of TiO_2_-NPs. However, no particles were seen to move paracellularly through the TJs. Rather, TiO_2_-NPs were internalized through ingestion and appeared later in the vacuoles.

### 2.3. Cultured Human Airway Epithelial (Calu-3) Cells

The Calu-3 is a mucus-secreting cell line derived from a pulmonary adenocarcinoma patient [61]. Calu-3 cells possess TJ structures [62] and yield the highest TEER values compared to other bronchial epithelial cell lines like 16HBE, H292, and BEAS-2B [63]. Several studies show that nanoparticle exposure to Calu-3 monolayers does not affect barrier function, cell viability, or cytotoxicity. A study by Braakhuis et al. measured the effect of 500 μg/mL of TiO_2_-NPs on Calu-3 cells [64]. The authors saw no changes in TEER or mitochondrial activity during the 24 h exposure period. Similarly, Stuetz et al. exposed Calu-3 cells to 0.25 mM and 2 mM of TiO_2_-NPs and ZnO-NPs for 24 h [65]. They showed that only ZnO-NP exposure caused a decrease in cell viability and TEER at all concentrations. On the other hand, cytotoxicity was not significantly affected by either TiO_2_-NPs or ZnO-NPs. Previous findings of Calu-3 exposure to SiO_2_-NPs are not uniform either. For example, George et al. exposed Calu-3 to 5 and 10 μg/cm^2^ of SiO_2_-NPs and TiO_2_-NPs for 24 h [66]. In this study, SiO_2_-NPs and TiO_2_-NPs were translocated and internalized across Calu-3 monolayers without damaging epithelial integrity. The authors observed an increase in TEER rather than the usual decrease that reflects diminished barrier function. In contrast, a study by McCarthy et al. assessed the impact of different-sized SiO_2_-NPs on Calu-3 cells [67]. They used 10 nm, 150 nm, and 500 nm-sized SiO_2_-NPs at a concentration of 100 μg/mL for a 24 h exposure period. The 10 nm-sized SiO_2_-NPs were found to induce cytotoxicity, apoptosis, and intracellular ROS generation in the Calu-3. However, SiO_2_-NPs that were 150 and 500 nm did not have any toxic effects. These data suggest that Calu-3 cells respond differently depending on the type and composition of nanoparticles.

While many other lung epithelial cell lines, such as H441 and BEAS-2B, have been used in cell culture studies, we have decided to focus on 16HBE, A549, and Calu-3 cell lines. These are three of the most frequently used airway epithelial cell lines in nanoparticle toxicity studies. The Calu-3 and 16HBE are advantageous because they exhibit tight junctions [18]. Moreover, all three cell lines can be used to study drug metabolism in the airway epithelium [68,69].

### 2.4. Human Bronchial Epithelial (NHBE) Cells

In addition to immortalized cells, primary cells are beneficial because they are more physiologically representative of the epithelial airway [70]. The challenges of using primary cells include their short lifespan, difficulty in isolating cells, and variability between cell samples [71]. However, normal human bronchial epithelial (NHBE) cells have well-established TJs [72] and are frequently used to study respiratory diseases like influenza and RSV [73,74,75].

The use of air–liquid interface (ALI) models to assess the toxicity of nanoparticles has emerged as an alternative or complementary method to traditional in vitro cell culture techniques [76]. For instance, Jing et al. investigated the toxicity of CuO-NPs using human bronchial epithelial cells (HBECs) and A549 cells in an ALI system [77]. Their study found that exposure to CuO-NPs led to a significant decrease in cell viability and an increase in oxidative stress, cytotoxicity, and IL-8 release. The authors show that their ALI system data aligned with findings from other murine studies on nanoparticle toxicity. Thus, ALI models are advantageous because they mimic in vivo conditions, unlike the submerged exposure of traditional cell culture models, which do not accurately represent real-life inhalation scenarios [78]. A notable benefit of ALI systems is the ability to use aerosol exposure, which allows for the more precise determination of nanoparticle dosages [79]. Although relatively new, ALI models show promise for future in vitro research on the impact of various nanoparticle types on the airway epithelial barrier.

Other studies have used NHBE cells in conjunction with an immortalized cell line to compare the toxic effects of nanoparticles in each cell type. Kim et al. exposed NHBE and A549 cells to SiO_2_-NPs in order to determine cytotoxicity, method of cell death, and intracellular NP accumulation [80]. NHBE and A549 cells were exposed to 10–400 µg/mL of SiO_2_-NPs for 4 h. Cell viability was seen to decrease significantly at 100 µg/mL for both cell lines. SiO_2_-NPs were found to induce necrosis in NHBE and A549 as well. Flow cytometry was used to measure the retention of FITC-labeled SiO_2_-NPs. The authors found that SiO_2_-NPs were contained in 95.62% of NHBE cells and 99.47% of A549 cells after exposure. In another study by Frontiñan-Rubio et al., the authors used NHBE and A549 cells to assess the toxic effects of GONPs [81]. NHBE exposure to 5 µg/mL of GONPs for 6 h was seen to significantly increase necrosis and apoptosis. Comparatively, A549 exposure to 5 µg/mL of GONPs for 24 h was seen to increase necrotic and apoptotic cells to a lesser degree. They found a slight, nonsignificant decrease in cell viability for 24 h exposure to 5 µg/mL of GONPs, while A549 cells remained unchanged. Oxidative stress was also examined by measuring hydrogen peroxide (H_2_O_2_) and superoxide anion (O_2_^−^) levels. There was a 51.3% increase in H_2_O_2_ for NHBE exposed to 5 µg/mL of GONPs for 24 h. However, O_2_^−^ levels remained the same after NHBE exposure to GONPs. No changes in H_2_O_2_ or O_2_^−^ levels were observed in A549 cells. These results show that A549 cells are more resistant to GONPs than the NHBE. On another note, Hussain et al. exposed NHBE and 16HBE cell lines to 20 μg/cm^2^ of TiO_2_-NPs for 4 h [82]. They revealed that TiO_2_-NPs caused intracellular ROS production in both NHBE and 16HBE cells after 30 min of exposure. DNA fragmentation was also seen in the cells, which confirms the occurrence of apoptosis. They found that TiO_2_-NPs induce apoptosis by destabilizing lysosomal membranes. Specifically, the proteases released by the damaged membrane have a significant role in apoptosis. Destabilization of the lysosomal membrane was observed after 30 min of exposure to TiO_2_-NPs. These findings are corroborated by Smallcombe et al., in which the authors exposed NHBE and 16HBE to 10–100 μg/mL of TiO_2_-NPs [40]. They showed that TiO_2_-NPs caused barrier disruption and AJC disassembly. For 16HBE, there was a significant decrease in TEER for 25–100 μg/mL of TiO_2_-NPs. Likewise, NHBE exposure to 25–100 μg/mL of TiO_2_-NPs induced a significant decrease in TEER. This indicates that exposure to TiO_2_-NPs increases the permeability of cell monolayers, which is in accordance with past studies on AEC exposure to TiO_2_-NPs. Immunolabeling of TJ and AJ proteins demonstrated that exposure to TiO_2_-NPs damaged the normal “chicken wire” strands seen in the control cells. They observed more gaps and decreased labeling intensity in monolayers exposed to TiO_2_-NPs. Furthermore, this study examined the effects of TiO_2_-NPs on RSV-infected 16HBE cells. In the presence of RSV, TiO_2_-NPs amplified RSV infection and its damaging effects on the AJC. This reflects the role of nanoparticles in exacerbating the harmful effects of pre-existing pulmonary conditions. Previous studies by Chakraborty et al. have also shown that TiO_2_-NPs induce apoptosis and inflammation. In one investigation, Chakraborty et al. exposed NHBE cells to 10 µg/mL of TiO_2_-NPs for 24 h before infection with the RFP-expressing respiratory syncytial virus (rrRSV) [83]. The authors found that pre-exposure of NHBE cells to TiO_2_-NPs increased their susceptibility to rrRSV infection by upregulating the expression of the nerve growth factor (NGF). Similarly, in another study, Chakraborty et al. exposed NHBE cells to 10 µg/mL of TiO_2_-NPs for 24 h and assessed the expression of various neurotrophic factors [84]. They observed an increase in NGF and p75^NRF^, an NGF receptor, as well as enhanced JNK phosphorylation, which promotes apoptosis. Additionally, these elevated neurotrophins caused by exposure to TiO_2_-NPs are believed to induce airway inflammation and hyperreactivity.

The impact of nanoparticles on primary and immortalized cell lines is generally concurring. However, differences in pro-inflammatory responses between NHBE and immortalized epithelial cells have also been published. Notably, a study by Ekstrand-Hammarström et al. compared cellular responses of NHBE and A549 cells that were exposed to TiO_2_-NPs and found that both cell lines showed similar changes in cell viability and oxidative stress, while cytokine production differed [85]. The investigators used five different TiO_2_-NPs, characterized by size and photocatalytic activity (anatase or rutile): 9 nm rutile (R9), 5 nm rutile (R5), 14 nm anatase (A14), 60 nm anatase (A60), and 20 nm mixed anatase and rutile (P25). First, both cell lines were exposed to 5–200 μg/mL of TiO_2_-NPs for 24 h, and ROS production was measured at 2 and 24 h time points. They showed that P25 and rutile TiO_2_-NPs caused significant ROS production in NHBE and A549 at both 2 and 24 h. To measure cell viability, NHBE and A549 were exposed to 0.19–400 μg/mL of TiO_2_-NPs for 24 h. There was a nonsignificant decrease in cell viability for both NHBE and A549 at 400 μg/mL. Following this, they evaluated the pro-inflammatory responses of the cell lines by measuring cytokine secretion. NHBE and A549 were exposed to 10, 50, and 250 μg/mL of TiO_2_-NPs for 24 h. There was a significant increase in IL-8, a pro-inflammatory cytokine, and secretion at 50 μg/mL of P25 for both NHBE and A549. Moreover, at exposure to 100 μg/mL of P25, there was a strong expression of cytokines IL-1β, VEGF, and G-SCF for NHBE cells that were not seen in the A549. This clarifies that nanoparticle-induced cytokine release varies with the cell culture model and composition of nanoparticles that are used.

Similarly, a study by Schlinkert et al. compared NHBE and A549 cells by exposing them to 0.1–0.8 μg/cm^2^ of AgNPs and AuNPs for 24 h [86]. No significant cytotoxicity was observed in A549 cells. On the other hand, 0.4 and 0.8 μg/cm^2^ of AgNPs caused a significant increase in LDH release for NHBE, showing cytotoxicity in higher nanoparticle concentrations. Cell viability was seen to decrease substantially in NHBE exposed to 0.7–0.8 μg/cm^2^ of AgNPs and AuNPs, while no changes were seen in A549 cells. Remarkably, ROS production was seen to be the most minimal in NHBE compared to the A549. This result contradicts Frontiñan-Rubio et al.’s findings [81] that NHBE cells are more sensitive to GONPs than A549 cells in terms of oxidative stress. However, considering the difference in properties of GONPs and AgNPs, it is evident that the cellular responses and susceptibility of each cell line depend on the type of nanoparticle to which it is exposed.

**Table 1 ijms-25-07885-t001:** Nanoparticle impact on AJC barrier and cellular responses using cell culture models.

Cell Culture Model	Nanoparticle Type	Nanoparticle Concentration	Impact on Barrier Function	Cellular Responses	Reference
16HBE14o- human bronchial epithelial cells (16HBE)	TiO_2_-NPs	10, 25, 50, 75, and 100 µg/mL	↓ TEER ↑ Permeability	↑ ROS production ↑ Viral infection ↑ Interleukins, TNF-α, IFN-γ, CCL-3, GM-CSF release	[40]
		100 µg/mL	↓ TEER ↑ Permeability	-	[41]
		0.1, 1, 10, and 100 µg/mL	-	↑ ROS production ↑ Cytotoxicity ↓ Cell viability ↓ DNA methylation	[42]
		3, 15, and 75 µg/cm^2^	-	↑ ROS production ↑ GM-CSF, IL-6, and IL-8 release ↑ Cytotoxicity	[43]
		20 µg/cm^2^	-	↑ ROS production ↑ Apoptosis ↑ DNA fragmentation ↓ Lysosome stability	[82]
	SiO_2_-NPs	40 µg/mL	-	↓ Cell viability ↓ Cell migration ↑ Apoptosis ↑ Oxidative stress ↑ IL-6, IL-8, TNF-α release	[44]
		3, 15, and 75 µg/cm^2^	-	↑ ROS production ↑ GM-CSF, IL-6, and IL-8 release ↑ Cytotoxicity	[43]
	ZnO-NPs	1, 5, 25, and 50 µg/mL	-	↑ ROS production ↓ Mitochondrial membrane potential ↑ Apoptosis	[45]
Human A549 adenocarcinoma cells (A549)	TiO_2_-NPs	40 µg/mL	-	↑ Endocytosis of aggregates	[60]
		1, 10, and 50 µg/mL	-	↑ ROS production ↓ Cell viability ↑ Cytotoxicity ↑ Apoptosis	[50]
		25, 50, 100, and 200 µg/mL	-	↑ Cytotoxicity ↑ Apoptosis ↑ DNA strand breaks ↓ Cell viability ↓ Mitochondrial membrane potential	[51]
		0.1, 1, 10, and 100 µg/mL	-	↑ ROS production ↑ Cytotoxicity ↓ Cell viability ↓ DNA methylation	[42]
		3, 15, and 75 µg/cm^2^	-	↑ ROS production ↑ IL-6 release ↑ Cytotoxicity	[43]
		5–500 µg/mL	-	↑ ROS production ↑ IL-8 release ↓ Cell viability	[85]
	AgNPs	25, 50, 100, or 200 µg/mL	-	↑ ROS production ↓ Cell viability ↓ Mitochondrial membrane potential ↑ Apoptosis	[52]
		5 µg/mL Co-culture with U937	-	↑ IL-1β, IL-6, IL-8, TNF-α release ↑ ROS production (less than A549 alone)	[58]
		0.05–0.8 µg/cm^2^	No change	↑ ROS production	[86]
	AuNPs	0.05–0.8 µg/cm^2^	No change	↑ ROS production	[86]
	SiO_2_-NPs	10, 50, 100, and 250 µg/mL	-	↓ Cell viability ↑ ROS production ↓ Mitochondrial membrane potential	[54]
		3, 15, and 75 µg/cm^2^	-	↑ ROS production ↑ GM-CSF, IL-6, IL-1β, IL-8 release ↑ Cytotoxicity	[43]
		10, 50, 100, 200, 300, 400 µg/mL	-	↓ Cell viability ↑ Necrosis ↑ NP accumulation and retention	[80]
	ZnO-NPs	5, 10, 15, 20 µg/mL	-	↓ Cell viability ↑ ROS production ↓ Mitochondrial membrane potential	[53]
	GONPs	0.05, 0.5, 5, 50, 100 µg/mL	-	↑ Necrosis ↑ Apoptosis ↓ Cell viability	[81]
	CaSO_4_-NPs	10–320 µg/mL	-	↑ ROS production	[59]
	ZnSO_4_-NPs	10–320 µg/mL	-	↑ ROS production ↑ DNA damage ↑ Cytotoxicity ↑ JNK regulation	
	PbSO_4_-NPs	10–320 µg/mL	-	↑ ROS production	
Cultured human airway epithelial cells (Calu-3)	TiO_2_-NPs	1 µg/cm^2^	No change	No changes in cell viability, barrier integrity, or cytokine release	[64]
		0.25–2 mM	No change	No changes in cell viability or cytotoxicity	[65]
		5 and 10 µg/cm^2^	↑ TEER	↑ NP internalization	[66]
	SiO_2_-NPs	5 and 10 µg/cm^2^	↑ TEER	↑ NP internalization	[66]
		100 µg/mL	-	↑ ROS production ↑ Cytotoxicity ↑ Apoptosis ↑ IL-6, IL-8 release	[67]
	ZnO-NPs	0.25–2 mM	↓ TEER	↓ Cell viability	[65]
Normal human bronchial epithelial cells (NHBE)	CuO-NPs	Air–liquid interface	-	↑ Oxidative stress ↑ Cytotoxicity ↑ IL-8 release ↓ Cell viability	[77]
	TiO_2_-NPs	20 µg/cm^2^	-	↑ ROS production ↑ Apoptosis ↑ DNA fragmentation ↓ Lysosome stability	[82]
		10, 25, 50, 75, and 100 µg/mL	↓ TEER ↑ Permeability	-	[40]
		5–500 µg/mL	-	↑ ROS production (weak) ↑ IL-6, IL-1β, G-CSF, VEGF release ↓ Cell viability	[85]
		10 µg/mL	-	↑ Susceptibility to rrRSV ↑ NGF expression	[83]
				↑ NGF expression ↑ p75^NRF^ receptor expression ↑ JNK phosphorylation	[84]
	SiO_2_-NPs	10, 50, 100, 200, 300, 400 µg/mL	-	↓ Cell viability ↑ Necrosis ↑ NP accumulation and retention	[80]
	GONPs	0.05, 0.5, 5, 50, 100 µg/mL	-	↑ Necrosis ↑ Apoptosis ↓ Cell viability ↑ Oxidative stress	[81]
	AgNPs	0.05–0.8 µg/cm^2^	-	↑ LDH release ↓ Cell viability ↑ ROS production (weak)	[86]
	AuNPs	0.05–0.8 µg/cm^2^	-	↑ LDH release ↓ Cell viability ↑ ROS production (weak)	[86]

List of abbreviations: AgNPs, silver nanoparticles; AuNPs, gold nanoparticles; CaSO_4_-NPs, calcium sulfate nanoparticles; CCL-3, macrophage inflammatory protein 1-α; G-CSF, granulocyte colony stimulating factor; GM-CSF, granulocyte macrophage colony stimulating factor; GONPs, graphene oxide nanoparticles; IL-1β, interleukin-1 beta; IL-6, interleukin-6; IL-8, interleukin-8; IFN-γ, interferon gamma; JNK, jun N-terminal kinase; LDH, lactate dehydrogenase; PbSO_4_-NPs, lead sulfate nanoparticles; NGF, nerve growth factor; ROS, reactive oxygen species; SiO_2_-NPs, silicon dioxide nanoparticles; TiO_2_-NPs, titanium dioxide nanoparticles; TNF-α, tumor necrosis factor alpha; VEGF, vascular endothelial growth factor; ZnO-NPs, zinc oxide nanoparticles; ZnSO_4_-NPs, zinc sulfate nanoparticles; ↓, decreased; ↑, increased; and -, not determined.

To summarize, both immortalized and primary cell culture models are extensively used to study the effects of metal and nonmetal nanoparticles on the respiratory tract. In vitro methods are cost-efficient, easily reproducible, and allow for controlled experiment conditions (Figure 2). Notably, NHBE cells closely resemble the bronchial epithelium seen in in vivo models. On the other hand, immortalized cell lines lack the specific characteristics of native epithelium, making it difficult to translate cell culture results to human responses. Our analysis indicates that 16HBE cells are most frequently used to investigate nanoparticle toxicity and its effects on the airway epithelial barrier. The 16HBE cell line is particularly advantageous due to its intact TJ structures and well-established TEER, both of which are crucial in evaluating barrier function.

## 3. Animal Models of Nanoparticle Toxicity on the Airway

Animal models are useful in assessing the impact of nanoparticle toxicity on the airway barrier because of their ability to manipulate the environment in a way that mimics real-life conditions. Specifically, many mouse and rat studies have looked at the effect of nanoparticles via inhalation on lung inflammation and other physiological responses [87,88,89] (Table 2). Moreover, animal models with pre-existing conditions are frequently used to study the impact of nanoparticle toxicity on the respiratory tract. This is beneficial because nanoparticle retention and clearance may appear to be different in the presence of chronic diseases. Using murine models can shed light on the mechanism by which nanoparticles disrupt lung function, cause inflammation, and aggravate underlying pulmonary diseases.

### 3.1. Mouse Models

A study by Alqahtani et al. examined the impact of AgNPs on mice exhibiting metabolic syndrome (MetS), which includes conditions like high cholesterol, hypertension, and insulin resistance [90]. Male C57BL/6J mice were fed either a regular or high-fat Western diet (HFWD) to increase cholesterol and body weight for 14 weeks. The mice were exposed to 1 mg/mL of AgNPs via oropharyngeal aspiration, and acute toxicity was measured 24 h after exposure. The authors found that AgNP exposure significantly increased the mRNA expression of pro-inflammatory mediators MIP-2, MCP-1, IL-6, IL-1β, and CXCL1 in both control and MetS mice. MetS mice exhibited even higher enhanced levels of MIP-2, IL-6, and MCP-1 compared to the healthy mice. Additionally, bronchoalveolar lavage fluid (BALF) analysis demonstrated that total cell and neutrophil counts were higher in both control and MetS mice exposed to AgNPs. Together, these responses provide evidence that AgNPs induce lung inflammation, and the presence of MetS exacerbates this effect.

Similarly, animal studies show that ZnO-NPs cause inflammatory responses in the lung, although at lower toxicity levels than AgNPs. For example, Adamcakova-Dodd et al. examined the toxic effects in mice after sub-acute or sub-chronic exposure to ZnO-NPs [91]. They exposed male C57BL/6 mice to 3.5 mg/m^3^ of ZnO-NPs for 4 h a day for a duration of 2 or 13 weeks to mimic sub-acute and sub-chronic conditions, respectively. There was an increased amount of Zn^2+^ ions in the BALF right after exposure to ZnO-NPs, but this reverted back to baseline concentrations after 3 weeks following exposure. In the mice exposed for 2 weeks, the authors found a significant increase in macrophages and a nonsignificant increase in IL-12(p40) and MIP-1α, which are inflammatory cytokines. However, the parameters for measuring lung toxicity did not differ significantly for the sub-chronic mice. This study reveals the relatively low toxicity of ZnO-NPs in murine models, especially for long-term inhalation exposure.

Evidence has shown that TiO_2_-NPs have different effects on murine models based on the type of disease present in the animal. Smallcombe et al. exposed female C57BL/6 mice to RSV and approximately 0.5–5 mg/kg of TiO_2_-NPs via intranasal instillation [40]. They collected BALF 4 days after RSV infection. The leukocyte count was significantly greater at concentrations of 2 and 3 mg/kg of TiO_2_-NPs, indicating inflammation of the airway barrier. Total BALF protein levels were elevated for all concentrations as well, showing increased permeability and characteristic of a disrupted airway barrier. In addition, hematoxylin and eosin (H&E) staining of lung tissue demonstrated a thickening of the airway wall and increased immune cells. This result further confirms the dose-dependent inflammation response induced by TiO_2_-NPs and RSV, both separately and together. Intriguingly, TiO_2_-NPs have been found to reduce pulmonary inflammation in asthmatic mice. Rossi et al. exposed female BALB/c/Sca mice to 10 ± 2 mg/m^3^ of TiO_2_-NPs three times a week for a total of four weeks [88]. The mice in the asthmatic condition were given 20 μg of ovalbumin intraperitoneally. The authors observed that healthy mice showed a significant increase in CXCL5, a chemokine that modulates leukocyte activity. However, asthmatic mice showed decreased levels of pro-inflammatory cytokines and chemokines. This points to how nanoparticles can suppress, rather than antagonize, the immune responses of certain respiratory conditions. Different-sized nanoparticles can also elicit varying inflammatory responses. In a study by Grassian et al., male C57BL/6 mice were exposed to TiO_2_-NPs ranging from 5 to 21 nm in size through either a whole-body exposure chamber or nasal instillation for 4 h and necropsied immediately or 24 h after the exposure [92]. BALF was utilized to quantify the total protein amount, LDH activity, and concentrations of cytokines IL-1β, IL-6, and TNF-α. The researchers discovered that the larger 21 nm TiO_2_-NPs induced greater inflammatory responses compared to the 5 nm TiO_2_-NPs. The BALF of mice exposed to the 21 nm TiO_2_-NPs exhibited a significantly increased number of neutrophils and LDH activity, which were not observed in those exposed to the 5 nm TiO_2_-NPs. Additionally, the larger TiO_2_-NPs prompted elevated concentrations of IL-1β and IL-6, while no change in TNF-α levels was noted. The authors concluded that the TiO_2_-NPs were delivered to the mice as agglomerates rather than individual particles, suggesting that the packing of nanoparticles could decrease the total surface area that is exposed to the mice, thus reducing nanoparticle toxicity. Therefore, the agglomeration state is an important factor when assessing nanoparticle toxicity in vivo.

Previous studies have demonstrated that cerium oxide nanoparticles (CeO_2_-NPs) induce inflammation and oxidative stress in mouse models. CeO_2_-NPs, commonly used as a diesel fuel additive, can lead to intracellular ROS production due to their oxidizing abilities [93]. Aalapati et al. exposed male CD1 mice to 2 mg/m^3^ of CeO_2_-NPs via inhalation for 6 h per day for 0, 7, 14, and 28 days [94]. BALF was collected 1 day after exposure and was analyzed for inflammatory responses. This study revealed a time-dependent decrease in cell viability over the 28-day exposure period. The neutrophil count in the BALF was significantly higher in mice exposed to CeO_2_-NPs, an indication of pulmonary inflammation. Furthermore, pro-inflammatory cytokines TNF-α, IL-1β, and IL-6 significantly increased, also in a time-dependent manner. Increased protein levels in the BALF of mice exposed for 28 days indicated decreased integrity of the airway epithelial barrier. A significant decrease in glutathione (GSH) levels in mice exposed to CeO_2_-NPs supported the occurrence of oxidative stress in epithelial cells. Histological analysis showed damage to the airway barrier, manifested as necrosis, fibrosis, proteinosis, and apparent granulomas in the pulmonary parenchyma. Similarly, Nemmar et al. investigated the impact of acute exposure to CeO_2_-NPs on the lungs [95]. Male and female BALB/c mice were exposed to 0.1 or 0.5 mg/kg of CeO_2_-NPs via intratracheal instillation. BALF collected 24 h after the initial delivery of CeO_2_-NPs showed significantly elevated total cell and neutrophil counts at both concentrations. CeO_2_-NPs induced an increase in TNF-α, an inflammatory cytokine, and a decrease in catalase activity, an antioxidant enzyme, in the BALF of exposed mice. Correspondingly, lung sections from mice showed that exposure to CeO_2_-NPs led to a dose-dependent increase in neutrophils and macrophages in the alveolar interstitial space. This study further demonstrates the oxidative stress and inflammation of the lungs caused by CeO_2_-NPs.

A comparison study of ZnO-NPs, TiO_2_-NPs, Al_2_O_3_-NPs, and CeO_2_-NPs was conducted by Larsen et al. to examine the acute and persistent impacts of each nanoparticle type [96]. Female BALB/cJ mice were exposed to an average of 4 to 271 mg/m^3^ of ZnO-NPs, TiO_2_-NPs, Al_2_O_3_-NPs, and CeO_2_-NPs via inhalation for 60 min. Lung inflammation was observed in the BALF of mice exposed to ZnO-NPs 24 h post-exposure, as there was a significant increase in neutrophils and lymphocytes. However, this increase was not observed in TiO_2_-NPs or Al_2_O_3_-NPs. Mice exposed to CeO_2_-NPs showed elevated neutrophil and lymphocyte levels at 13 weeks post-exposure. Interestingly, only TiO_2_-NPs showed a significant increase in DNA-strand breaks, indicating damage to DNA. The amount of break time taken during the inhalation period by each mouse was used to determine their nose irritation response. A significantly higher break time was observed in mice exposed to TiO_2_-NPs and ZnO-NPs but not for Al_2_O_3_-NPs or CeO_2_-NPs. The authors found that ZnO-NPs induced more intense and persistent damaging effects than the other nanoparticles. Overall, TiO_2_-NPs and Al_2_O_3_-NPs were ranked as having low inflammatory responses and potency, while CeO_2_-NPs fell in between ZnO-NPs and TiO_2_-NPs/Al_2_O_3_-NPs.

Like other transition metal nanoparticles, CoNPs and CuO-NPs have been shown to induce oxidative stress, inflammation, and DNA damage. For instance, Wan et al. exposed male and female *gpt* delta transgenic mice to 50 μg of CoNPs via intratracheal instillation [97]. BALF was collected at time points 1, 3, 7, and 28 days, or four months after the initial delivery of CoNPs. The authors observed an increase in neutrophils, LDH activity, total protein levels, and the amount of chemokine CXCL1/KC in the BALF of mice exposed to CoNPs. Histopathological analysis showed that neutrophils and macrophages infiltrated the alveolar space and interstitial tissues 7 days after the instillation of CoNPs. The alveolar wall was also thickened. These findings indicate lung injury and inflammation. Immunohistochemical staining of Ki-67 and PCNA, indicators of cell proliferation, and γ-H2AX, which measures DNA damage, was performed. Ki-67 and PCNA were confirmed to be in the nucleus, and the number of Ki-67- and PCNA-positive cells significantly increased. Even at the 4-month mark, CoNPs were found to elevate Ki-67 and PCNA levels. Furthermore, CoNPs induced DNA damage, shown by the higher frequency of transversion mutations and 8-OHdG levels in the lung tissue.

Similarly, Lai et al. investigated the impact of CuO-NPs on C57BL/6 mice [98]. They exposed the mice to 2.5 mg/kg, 5 mg/kg, and 10 mg/kg of CuO-NPs via intranasal instillation, and inflammation parameters were assessed at 7, 14, and 28 days after the initial exposure. Hematoxylin and eosin (H&E) staining of lung tissue collected at the 14-day mark showed inflammation. This was supported by a significant increase in pro-inflammatory gene expression such as CCL-2, IL-4, and TNF-α for 5 mg/kg of CuO-NPs. The authors also observed the expression of α-SMA, a marker for myofibroblast activation, and collagen-I in mice exposed to CuO-NPs, indicating that CuO-NPs induce lung fibrosis in addition to inflammation. Likewise, Pietrofesa et al. exposed female C57BL/6 mice to 15 µg/bolus of CuO-NPs via intranasal instillation, and BALF was collected at 1, 3, and 7 days after instillation [99]. At the 1-day time point, a significant increase in leukocyte, total protein, and neutrophil count was observed in the BALF of mice exposed to CuO-NPs. The authors found a significant elevation in protein chlorination caused by the influx of neutrophils and macrophages abundant in myeloperoxidase (MPO), which release HOCl when inflammation occurs. Pro-inflammatory cytokine levels of HMGB1, IL-1β, and TNF-α were significantly increased as well. These data show how CuO-NPs lead to pulmonary injury and inflammation in the murine lung.

### 3.2. Rat Models

Like mice, rats are another species used for nanoparticle toxicity studies. In one example, Braakhuis et al. exposed male F344/DuCrl rats to AgNPs ranging from 41 to 1105 mg/m^3^ of air for 4 days [100]. They analyzed the BALF and found a concentration-dependent increase in the number of cells, neutrophils, and pro-inflammatory markers IL-1β and MCP-1. The authors observed an increase in the total lung deposition of nanoparticles as the concentration of AgNP exposure increased. The total amount of AgNPs in the lungs decreased for all concentrations 7 days after the initial exposure. This study’s findings indicate how AgNPs lead to pulmonary toxicity and inflammation.

TiO_2_-NPs have been shown to cause changes in neurotrophins, leading to airway inflammation and hyperresponsiveness. For instance, Scuri et al. exposed 1- to 2-day-old, 2-week-old, and 12-week-old rats to 12 mg/m^3^ of TiO_2_-NPs via inhalation for 5.6 h daily for 3 consecutive days [101]. They observed an increase in the nerve growth factor (NGF) and brain-derived neurotrophic factor (BDNF) expression for 2-week-old rats, but only elevated NGF levels in 1- to 2-day-old rats. The 12-week-old rats did not exhibit any changes in NGF or BDNF expression. Following this, pro-inflammatory factors were assessed in the BALF of the rats. GRO/KC, a chemokine seen in inflammatory responses, was seen to increase in 2-week-old rats exposed to TiO_2_-NPs. This study demonstrated that exposure to TiO_2_-NPs impacts lung neurotrophins, which are associated with airway hyperreactivity and inflammation. These results are age-dependent, implying that nanoparticle exposure is more likely to cause substantial damage at the time of early lung development.

Morimoto et al. exposed male F344 rats to 0.8 mg/kg and 4 mg/kg of ZnO-NPs by intratracheal instillation, as well as 2 and 10 mg/m^3^ of ZnO-NPs via inhalation for 6 h every day [102]. In the BALF of rats exposed to ZnO-NPs via intratracheal instillation, there was a significant increase in the total cell and neutrophil count for all concentrations. On the other hand, rats that were exposed to ZnO-NPs via inhalation showed the same results only at the highest concentration. Furthermore, the expression of HO-1, an enzyme that is produced in response to oxidative stress, followed the same trend. Rats exposed to ZnO-NPs by intratracheal instillation had significantly higher HO-1 levels at all ZnO-NP concentrations, while this only held true at 10 mg/m^3^ of ZnO-NPs for rats that were exposed via inhalation. It is important to mention that these findings were not persistent. The BALF was analyzed at different time points ranging from 3 days to 6 months following the completion of instillation or inhalation exposure. Significant results were only seen at the 3-day mark, which shows that the toxic effects of ZnO-NPs are somewhat transient.

Rat models have shown similar outcomes to CeO_2_-NP exposure as seen in mice studies. Srinivas et al. examined the toxic effects of CeO_2_-NPs on male and female Wistar rats [103]. Rats were exposed to approximately 641 mg/m^3^ of CeO_2_-NPs via inhalation for 4 h, and BALF was collected at 24 h, 48 h, and 14 days following the exposure. The authors observed a significant decrease in cell viability and an increase in total cell count at all time points. A BALF analysis revealed significant increases in lactate dehydrogenase, total leukocyte count, and neutrophils. Rats exposed to CeO_2_-NPs showed an increase in pro-inflammatory cytokines IL-1β, TNF-α, and IL-6 in both BALF and blood samples, indicating the initiation of inflammatory responses. The presence of microgranulomas in the pulmonary parenchyma at the 14-day mark suggested a dysfunctional clearance mechanism resulting in CeO_2_-NP persistence in the lungs. Likewise, Demokritou et al. exposed male Sprague-Dawley rats to 2.7 mg/m^3^ of CeO_2_-NPs or CeO_2_-NPs coated in SiO_2_ for 2 h a day over 4 days via a whole-body inhalation chamber [104]. They found a significant increase in polymorphonuclear neutrophils (PMNs) and lactate dehydrogenase (LDH), markers for inflammation and cytotoxicity, respectively. A BALF analysis of rats exposed to SiO_2_-coated CeO_2_-NPs showed similar levels of albumin, PMNs, and LDH to those of healthy rats, indicating that encapsulation in SiO_2_ attenuated the adverse effects of CeO_2_-NPs. The authors highlighted the harmless nature of amorphous SiO_2_ as a reason why cells did not exhibit toxicity when exposed to CeO_2_-NPs coated by SiO_2_.

Past research has indicated that Al_2_O_3_-NPs are less toxic than other metal oxide nanoparticles [105]. However, murine studies suggest potential inflammatory effects of Al_2_O_3_-NPs on the respiratory system. In a study by Kim et al., male Sprague-Dawley rats were exposed to 0.2, 1, and 5 mg/m^3^ of Al_2_O_3_-NPs for 28 days via nasal inhalation only [106]. A BALF analysis showed a significantly higher count of total cells and neutrophils in the rats exposed to Al_2_O_3_-NPs at 1 and 5 mg/m^3^ of Al_2_O_3_-NPs. Lactate dehydrogenase, TNF-α, and IL-6 levels were also elevated at these concentrations. An examination of lung sections revealed alveolar macrophage accumulation in half of the rats exposed to 5 mg/m^3^ of Al_2_O_3_-NPs. Consistently, Yousef et al. compared the toxic impacts of Al_2_O_3_-NPs and ZnO-NPs [107]. Male Wistar rats were exposed to 70 mg/kg of Al_2_O_3_-NPs, 100 mg/kg of ZnO-NPs, or a combination of both daily via oral gavage for 75 days. They found that exposure to Al_2_O_3_-NPs and ZnO-NPs separately caused an increase in 8-hydroxy-2’-deoxyguanosine (8-OHdG), a biomarker for measuring oxidative damage to DNA, as well as cytokines TNF-α and IL-6. These results were accompanied by a decrease in GSH levels, indicating that both Al_2_O_3_-NPs and ZnO-NPs induced oxidative stress in lung tissues. The combination of Al_2_O_3_-NPs and ZnO-NPs exacerbated these results to a small degree.

Furthermore, oxidative stress and inflammation are key findings of rat studies using CoNPs and CuO-NPs. Hansen et al. investigated the effect of CoNPs on the formation of sarcomas using a rat model [108]. They bilaterally implanted 100 mg of CoNPs in Sprague–Dawley rats for 6, 8, or 12 months. At the 6-month time point, histological analysis found the presence of preneoplasia in three of the rats exposed to CoNPs. The authors observed enhanced nuclei and mitotic rates, as well as expression of PCNA in the mesenchymal cells of these rats. The findings for rats implanted with CoNPs show that initial inflammation led to preneoplasia, which eventually presents as neoplasia. This sequence is consistent with the pathogenesis of malignant tumors. Therefore, the findings of this study establish the role of CoNPs in advancing the process of neoplasia. It is worth noting that different Co-based nanoparticles elicit distinct responses. To highlight, Jeong et al. compared the effects of CoO-NPs and Co_3_O_4_-NPs on female rats [109]. The rats were exposed to 40, 100, and 400 μg/rat of CoO-NPs and Co_3_O_4_-NPs via intratracheal instillation. The BALF collected 24 h after instillation showed that LDH levels followed a dose-dependent significant increase for rats exposed to CoO-NPs. However, Co_3_O_4_-NPs only induced a significant elevation in LDH only at the highest dose, 400 μg/rat. Total protein concentration was significantly higher at 100, and 400 μg/rat of CoO-NPs, while rats exposed to Co_3_O_4_-NPs did not show any significant increases. The authors observed that Co_3_O_4_-NPs only induced an increase in cytokine-induced neutrophil chemoattractant-3 (CINC-3) levels. On the other hand, exposure to CoO-NPs led to greater levels of other pro-inflammatory cytokines such as IL-6, eotaxin, and IL-13. Additionally, two different types of inflammatory responses were shown. Rats instilled with CoO-NPs produced eosinophilic inflammation, while Co_3_O_4_-NPs led to neutrophilic inflammation. The contrast in cellular responses can be explained by the solubility properties of CoO-NPs and Co_3_O_4_-NPs. The investigators measured the solubility percentage of each nanoparticle type in artificial lysosomal fluid (ALF). They found that CoO-NPs had 92.65% solubility and Co_3_O_4_-NPs had 11.46% in ALF. Thus, the reported results support how varying physicochemical characteristics within the same element can enhance its inflammation and toxicity levels.

Additionally, Kwon et al. intratracheally instilled 0.15 and 1.5 mg/kg of CuO-NPs in male Sprague–Dawley rats [110]. An analysis of the BALF taken 24 h after the exposure showed a significant increase in the total cell and polymorphonuclear leukocyte (PMN) count in a dose-dependent manner. LDH activity and protein concentration in the BALF were also elevated compared to the healthy rats. Cytokine levels of MIP-2 and TNF-α were increased significantly in rats exposed to 1.5 mg/kg of CuO-NPs. Lung sections of rats exposed to CuO-NPs exhibited acute inflammation in the bronchioles and alveoli region. The highest concentration of CuO-NPs induced pulmonary edema in the lung tissue. Lastly, antioxidant expression of catalase, Gpx-1, and Prx-2 was down-regulated in the rats exposed to 1.5 mg/kg of CuO-NPs.

**Table 2 ijms-25-07885-t002:** Effects of nanoparticle exposure on murine models.

Murine Model	Nanoparticle Type	Concentration and Exposure Route	Inflammatory Response	BALF Analysis and Other Cellular Responses	Reference
Male C57BL/6J mice	AgNPs	1 mg/mL, oropharyngeal aspiration	↑ Expression of MIP-2, MCP-1, IL-6, IL-1β, CXCL1	↑ Total cell count ↑ Neutrophil count	[90]
Male C57BL/6 mice	ZnO-NPs	3.5 mg/m^3^, inhalation, 2 weeks	↑ Expression of IL-12p(40) and MIP-1α	↑ Zn^2+^ ions ↑ Total cell count ↑ Neutrophil count ↑ Macrophage count ↑ LDH release ↑ Hematocrit	[91]
		3.5 mg/m^3^, inhalation, 13 weeks	-	↑ Zn^2+^ ions ↑ Total cell count ↑ Macrophage count ↑ Hematocrit	
Female BALB/cJ mice		4–271 mg/m^3^, inhalation	-	↑ Neutrophil count ↑ Lymphocyte count ↑ Break time	[96]
Female C57BL/6 mice	TiO_2_-NPs	0.5–5 mg/kg, intranasal instillation	↑ Leukocyte count ↑ Expression of IL-1α, IL-6, IL-10, TNF-α, IFN-γ, MCP-1, RANTES, LIF, IP-10	↑ Total protein ↑ Barrier permeability	[40]
Female BALB/c/Sca mice		10 ± 2 mg/m^3^, inhalation, asthmatic mice	↓ Expression of IL-1β, TNF-α, IL-4, IL-13, IL-10, Foxp3 ↓ Expression of CCL-3, CXCL5, CXCL2 ↓ IgE levels	↓ Eosinophil count ↓ Lymphocyte count ↓ PAS+ goblet cells	[88]
		10 ± 2 mg/m^3^, inhalation, healthy mice	↓ Expression of IL-1β ↑ Expression of CXCL5	↑ Neutrophil count	
Male C57BL/6 mice		0.77–7.03 mg/m^3^, inhalation	↑ Expression of IL-1β and IL-6	↑ Total cell count ↑ Macrophage count	[92]
		0.1–3.0 mg/mL, intranasal instillation		↑ Total cell count ↑ Neutrophil count	
Female BALB/cJ mice		4–271 mg/m^3^, inhalation	-	↑ DNA damage ↑ Break time	[96]
Male CD1 mice	CeO_2_-NPs	2 mg/m^3^, inhalation	↑ Expression of TNF-α, IL-1β, and IL-6	↓ Cell viability ↑ Neutrophil count ↑ Total protein ↓ Glutathione (GSH)	[94]
Male and female BALB/c mice		0.1 or 0.5 mg/kg, intratracheal instillation	↑ Expression of TNF-α ↓ Catalase	↑ Total cell count ↑ Neutrophil count	[95]
Female BALB/cJ mice		4–271 mg/m^3^, inhalation	-	↑ Neutrophil count ↑ Lymphocyte count	[96]
Male and female gpt delta transgenic mice	CoNPs	50 μg, intratracheal instillation	↑ CXCL1/KC levels	↑ LDH release ↑ Neutrophil count ↑ Macrophage count ↑ Total protein ↑ DNA damage	[97]
C57BL/6 mice	CuO-NPs	2.5, 5, 10 mg/kg, intranasal instillation	↑ Expression of CCL-2, IL-4, TNF-α, α-SMA, and collagen-I	↑ Apoptosis	[98]
Female C57BL/6 mice		15 µg/bolus, intranasal instillation	↑ Leukocyte count ↑ HMGB1, IL-1β, and TNF-α levels	↑ Neutrophil count ↑ Protein chlorination ↑ Total protein	[99]
Male F344/DuCrl rats	AgNPs	41–1105 µg/m^3^, nose-only inhalation	↑ Expression of MCP-1, IL-1β	↑ Total cell count ↑ Neutrophil count ↑ LDH release ↑ Concentration-dependent lung deposition	[100]
Male and female Fischer 344 rats	TiO_2_-NPs	12 mg/m^3^, inhalation	↑ NGF and BDNF (Only seen in 2-week-old rats)	↑ GRO/KC (Only seen in 2-week-old rats)	[101]
Male Fischer 344 rats	ZnO-NPs	0.8 and 4 mg/kg, intratracheal instillation	↑ CINC-1 and CINC-2 (Only seen at highest concentration)	↑ Total cell count ↑ Neutrophil count ↑ LDH release ↑ Oxidative stress ↑ Macrophage count (Only seen at highest concentration)	[102]
		2 and 10 mg/m^3^, inhalation			
Male Wistar rats		100 mg/kg, oral gavage	↑ TNF-α and IL-6 levels	↑ 8-OHdG levels ↓ Glutathione (GSH)	[107]
Male and female Wistar rats	CeO_2_-NPs	641 mg/m^3^, inhalation	↑ Leukocyte count ↑ Expression of TNF-α, IL-1β, and IL-6	↓ Cell viability ↑ Total cell count ↑ LDH release ↑ Neutrophil count	[103]
Male Sprague-Dawley rats		2.7 mg/m^3^, inhalation	-	↑ LDH release ↑ PMNs	[104]
Male Sprague-Dawley rats	Al_2_O_3_-NPs	0.2, 1, and 5 mg/m^3^, nasal inhalation	↑ Expression of TNF-α and IL-6	↑ Total cell count ↑ LDH release	[106]
Male Wistar rats		70 mg/kg, oral gavage	↑ TNF-α and IL-6 levels	↑ 8-OHdG levels ↓ Glutathione (GSH)	[107]
Sprague-Dawley rats	CoNPs	100 mg, bilateral implantation	-	↑ Nuclei and mitotic rates ↑ Expression of PCNA	[108]
Female rats	CoO-NPs	40, 100, and 400 μg/rat, intratracheal instillation	↑ IL-6, eotaxin, and IL-13 levels ↑ Eosinophilic inflammation	↑ LDH release ↑ Total protein ↑ Solubility in ALF	[109]
	Co_3_O_4_-NPs		↑ CINC-3 levels ↑ Neutrophilic inflammation	↑ LDH release (only seen at highest concentration) ↓ Solubility in ALF	
Male Sprague-Dawley rats	CuO-NPs	0.15 and 1.5 mg/kg, intratracheal instillation	↑ MIP-2 and TNF-α levels	↑ Total cell count ↑ PMN count ↑ Total protein ↑ LDH release ↓ Expression of catalase, Gpx-1, and Prx-2	[110]

List of abbreviations: 8-OHdG, 8-hydroxy-2′-deoxyguanosine; α-SMA, alpha-smooth muscle actin; AgNPs, silver nanoparticles; ALF, artificial lysosomal fluid; Al_2_O_3_-NPs, aluminum oxide nanoparticles; BALF, bronchoalveolar lavage fluid; BDNF, brain-derived neurotrophic factor; CCL-2, chemokine (C-C motif) ligand 2; CCL-3, macrophage inflammatory protein 1-α; CeO_2_-NPs, cerium oxide nanoparticles; CINC-1, -2, -3, cytokine-induced neutrophil chemoattractant 1, 2, 3; CoNPs, cobalt nanoparticles; CoO-NPs, cobalt (II) oxide nanoparticles; Co_3_O_4_-NPs, cobalt (II, III) oxide nanoparticles; CuO-NPs, copper oxide nanoparticles; CXCL1, growth-regulated oncogene alpha; CXCL2, macrophage inflammatory protein-2; CXCL5, CXC chemokine ligand 5; Foxp3, forkhead box P3; GSH, glutathione; Gpx-1, glutathione peroxidase-1; HMGB1, high mobility group box 1; IFN-γ, interferon gamma; IgE, immunoglobulin E; IL-1α, -1β, -4, -6, -10, and -13, interleukin-1, alpha, beta, 4, 6, 10, and 13; IL-12p(40), interleukin-12 subunit p40; IP-10, interferon-gamma inducible protein of 10 kDa; LDH, lactate dehydrogenase; LIF, leukemia inhibitory factor; MCP-1, monocyte chemoattractant protein-1; MIP-1α and -2, macrophage inflammatory protein-1 alpha and 2; NGF, nerve growth factor; PAS, periodic acid-Schiff; PCNA, proliferating cell nuclear antigen; PMNs, polymorphonuclear neutrophils; Prx-2, peroxiredoxin-2; RANTES, regulated upon activation, normal T cell expressed, and secreted; TiO_2_-NPs, titanium dioxide nanoparticles; TNF-α, tumor necrosis factor alpha; ZnO-NPs, zinc oxide nanoparticles; ↓, decreased; ↑, increased; and -, not determined.

Taken together, mice and rat models are commonly used for in vivo studies to examine the impact of various nanoparticle types on the airway epithelial barrier, airway hyperreactivity, and inflammation. Mice, in particular, are favored due to their biological relevance and their suitability for studying chronic nanoparticle exposures via inhalation, something not feasible in traditional cell culture models (Figure 2). While the lung physiology of mice is not fully representative of human lungs, these murine models can still provide valuable data on how the function and structure of the airway epithelial barrier change after nanoparticle exposures, as well as how these exposures lead to detrimental changes to airway inflammation.

## 4. Conclusions and Future Direction

Nanoparticles, categorized as metal, metal oxide, or nonmetal, are extensively employed owing to their unique properties. However, exposure to nanoparticles via inhalation can cause detrimental effects on the respiratory tract. The airway epithelial barrier is a vital part of the human body’s innate immune system. TJs and AJs make up the AJC, which provide a barrier to inhaled pathogens and environmental particulate matter [111]. Markedly, the AJC facilitates the interactions between nanoparticles and the airway barrier. Previous research has shown that nanoparticles, especially those containing metal groups, disrupt the airway epithelial barrier and lead to increased barrier permeability, apoptosis, ROS production, DNA damage, and inflammation.

While there is expanding knowledge on the impact of different nanoparticle types on the airway epithelial barrier, there are limitations to current studies. Establishing a controlled environment that resembles real-world human exposure conditions is crucial but challenging for in vivo and in vitro experiments. To address this, utilizing combined exposure methods to multiple nanoparticle types or environmental pollutants will be more representative of the human surroundings. Also, extending the exposure time to look at the long-term effects of nanoparticles on the airway barrier can be beneficial for studying chronic respiratory outcomes. This prolonged exposure allows for greater insight into the progression of underlying pulmonary conditions and possible cumulative effects of pollutants. In addition, incorporating a wide range of nanoparticle concentrations can inform us of the dose-dependent relationship between nanoparticles and physiological responses. Furthermore, when comparing data from cell culture and murine studies, it is necessary to use standardized methods to ensure the reliability of the results.

Recent cell culture and murine model studies have focused on the cellular responses to different metal and nonmetal nanoparticles. Specifically, these investigations looked at airway epithelial barrier disruption, inflammatory reactions, and oxidative stress. The variation in nanoparticle composition plays a pivotal role in determining its impact on the respiratory tract. Understanding the specific effects of nanoparticle types on the airway epithelial barrier may advance the development of tailored experimental models and potential therapeutic interventions for nanoparticle-induced barrier disruption.

## Figures and Tables

**Figure 1 ijms-25-07885-f001:**
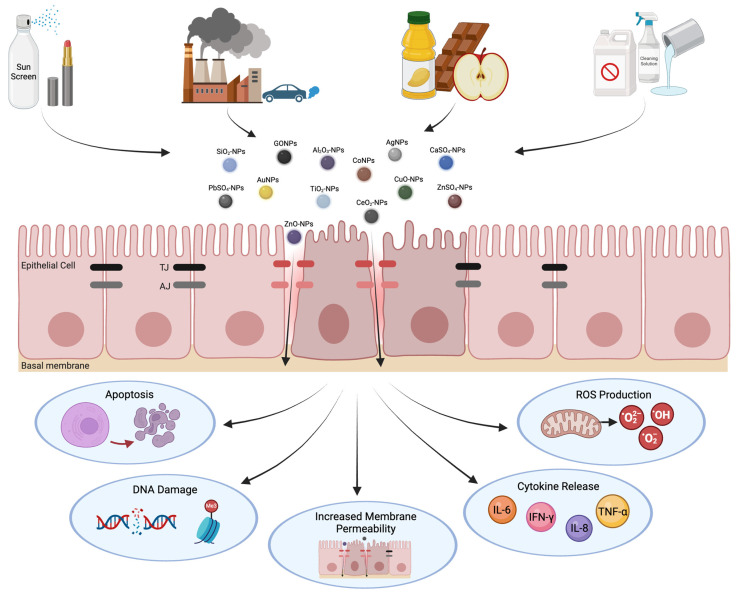
Adverse cellular responses to nanoparticle exposure. Inhalation of nanoparticles occurs through various applications, including cosmetics, cleaning products, occupational exposure, and food additives. Upon exposure, nanoparticles can penetrate the epithelial barrier, causing dysfunction of AJCs. The disruption of TJ and AJ structures leads to harmful cellular responses such as cell death, genetic alterations, inflammation, and oxidative stress. This figure was created with BioRender.com.

**Figure 2 ijms-25-07885-f002:**
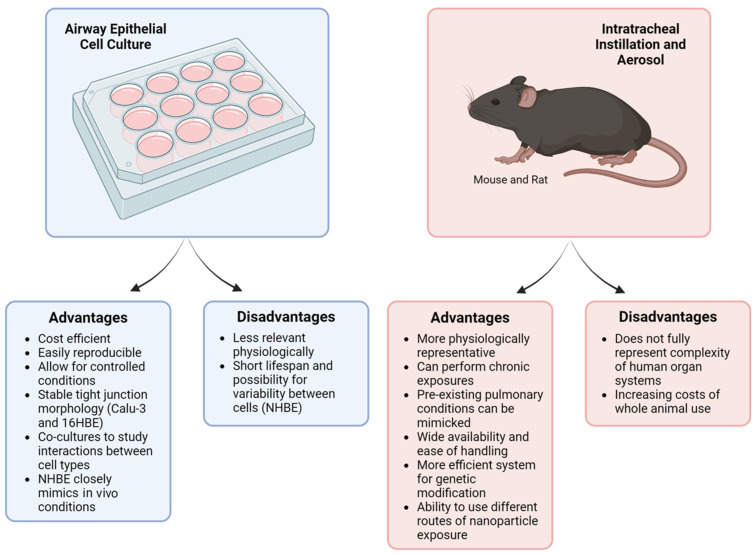
Comparison of cell culture and murine models. Cell culture and animal models have been utilized to investigate the impact of various nanoparticles on the lung. While each system offers distinct advantages, they also have notable limitations. This figure summarizes the key advantages and disadvantages of both cell culture and murine models. Created with BioRender.com.

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
