# Peer review of "Nanoparticles and Airway Epithelial Cells: Exploring the Impacts and Methodologies in Toxicity Assessment"

_ijms, 2024, doi:10.3390/ijms25147885_

Round 1

Reviewer 1 Report

Comments and Suggestions for Authors

Lee and Rezaee summarized current knowledge regarding the applications of in vitro and in vivo models for the identification of nanoparticle-induced cellular responses and barrier dysfunction.

This manuscript compiles 104 references, demonstrating that the authors have thoroughly researched the literature. However, there are some concerns.

As the authors aimed to analyze the advantages or disadvantages of different types of in vitro and in vivo models, it would be useful for the reader to add images summarizing the main features of the most used ones.

For the same reasons, a final summary highlighting the advantages of one model over another in the study of nanoparticle-induced toxicity might be useful to the reader.

Although the authors extensively analyzed the use of different cell lines, there has recently been a great focus on 3D models, such as air-liquid interface systems or organoids. A paragraph on the advantages of these models would be interesting (PMID: 25575782)

Finally, although the authors focused on alterations in the cell barrier, it is interesting to note that specific alterations caused by nanoparticles have been noted in the literature. For example, some authors have shown that exposure to titanium dioxide nanoparticles triggers neurogenic responses. As neurogenic inflammation is known to be an essential component in the development of bronchial hyperreactivity, it would be interesting to address the correlation between nanoparticle exposure and airway alterations (PMID: 20818535, PMID: 12505544).

This is an example of how different models can be used to identify nanoparticle-induced lung damage. 

Author Response

Lee and Rezaee summarized current knowledge regarding the applications of in vitro and in vivo models for the identification of nanoparticle-induced cellular responses and barrier dysfunction. This manuscript compiles 104 references, demonstrating that the authors have thoroughly researched the literature. However, there are some concerns. As the authors aimed to analyze the advantages or disadvantages of different types of in vitro and in vivo models, it would be useful for the reader to add images summarizing the main features of the most used ones.

Response: Thank you for your thoughtful comments. We have added an additional Figure 2 which illustrates the advantages and disadvantages of airway epithelial cell culture and mice models, showing the key features of each.

For the same reasons, a final summary highlighting the advantages of one model over another in the study of nanoparticle-induced toxicity might be useful to the reader.

Response: We appreciate your feedback and have added a paragraph at the end of both in vitro and in vivo sections summarizing the benefits of each model as well as the most common type used. This can be found in lines 398 to 407 and lines 680 to 688 in the new manuscript draft.

Although the authors extensively analyzed the use of different cell lines, there has recently been a great focus on 3D models, such as air-liquid interface systems or organoids. A paragraph on the advantages of these models would be interesting (PMID: 25575782)

Response: Thank you for this suggestion. We have added a paragraph on the advantages of air-liquid interface in vitro models with relevant citations in lines 299 to 311 in the revised manuscript.

Finally, although the authors focused on alterations in the cell barrier, it is interesting to note that specific alterations caused by nanoparticles have been noted in the literature. For example, some authors have shown that exposure to titanium dioxide nanoparticles triggers neurogenic responses. As neurogenic inflammation is known to be an essential component in the development of bronchial hyperreactivity, it would be interesting to address the correlation between nanoparticle exposure and airway alterations (PMID: 20818535, PMID: 12505544). This is an example of how different models can be used to identify nanoparticle-induced lung damage.

Response: Your insight and suggestions are greatly appreciated. We have included a paragraph to the animal model section detailing how exposure to TiO2-NPs cause neurotrophins to change, which correlate to airway inflammation and hyperresponsiveness, based on the study you have provided (PMID: 20818535). This can be found in lines 571 to 583 on the revised document.

We looked for PMID: 12505544 but could not find the relevant article on the impact of nanoparticle exposure on neurotrophins. Instead, we added two more articles to our manuscript (PMID: 28701524, PMID: 28140833) which talk about how NHBE exposure to TiO2-NPs induce apoptosis and susceptibility to RSV, both mediated through nerve growth factor (NGF). This can be found in lines 352 to 362 on the revised document.

Reviewer 2 Report

Comments and Suggestions for Authors

This is a nice article about cell and animal models used in the studies on the influence of nanoparticles on the respiratory tract. It summarizes many studies and gives a good picture of current research. There is one main issue I would like to address to the authors. Paragraph 2 is organized according to the cell line and paragraph 3 to the type of NPs. Consider to unify the approach, both paragraphs by models or by type of NPs. Maybe it would be an idea to summarize the best cell and animal model(s) for the research on NPs? Otherwise, explain NPs, GONPs and others when used for the first time and check titles and subtitles for typing errors. Was the dose of NPs in line 322 and others in mg per m^3 or per dm^3?

Author Response

This is a nice article about cell and animal models used in the studies on the influence of nanoparticles on the respiratory tract. It summarizes many studies and gives a good picture of current research. There is one main issue I would like to address to the authors. Paragraph 2 is organized according to the cell line and paragraph 3 to the type of NPs. Consider to unify the approach, both paragraphs by models or by type of NPs. Maybe it would be an idea to summarize the best cell and animal model(s) for the research on NPs?

Response: Thank you for your kind comments and we appreciate your valuable feedback. We have rearranged the animal model section by type of murine species to conform the subsections for both in vitro and in vivo models. We have changed Table 2 accordingly as well.

Otherwise, explain NPs, GONPs and others when used for the first time and check titles and subtitles for typing errors. Was the dose of NPs in line 322 and others in mg per m^3 or per dm^3?

Response: Thank you for bringing this to our attention. We wrote out the abbreviations when using the nanoparticle names for the first time and double checked the grammar in the manuscript. The dose in line 322 was 20 µg/mL as this was a study using cell culture.

Round 2

Reviewer 1 Report

Comments and Suggestions for Authors

I appreciate the efforts of the authors to address all comments raised previously, with additional figures and explanations. The manuscript has been improved. The overview on the applications of in vitro and in vivo models for the identification of nanoparticle-induced toxicity results clearer and more understandable. I believe that the scientific quality of the revised manuscript now corresponds to be published in International Journal of Molecular Sciences journal.

Reviewer 2 Report

Comments and Suggestions for Authors

I don't have any further comments.